# Identification of D-arabinan-degrading enzymes in mycobacteria

Omar Al-Jourani[1,11], Samuel T. Benedict[2,11], Jennifer Ross [1,11], Abigail J. Layton[2], Phillip van der Peet[3], Victoria M. Marando [4,5], Nicholas P. Bailey[1], Tiaan Heunis[1], Joseph Manion[1], Francesca Mensitieri[1], Aaron Franklin[2], Javier Abellon-Ruiz [1], Sophia L. Oram[1], Lauren Parsons[1], Alan Cartmell [6], Gareth S. A. Wright [7], Arnaud Baslé[1], Matthias Trost[1], Bernard Henrissat [8,9], Jose Munoz-Munoz [10], Robert P. Hirt [1], Laura L. Kiessling [4], Andrew L. Lovering [2], Spencer J. Williams [3], Elisabeth C. Lowe [1] ✉ & Patrick J. Moynihan [2] ✉

Bacterial cell growth and division require the coordinated action of enzymes that synthesize and degrade cell wall polymers. Here, we identify enzymes that cleave the D-arabinan core of arabinogalactan, an unusual component of the cell wall of *Mycobacterium tuberculosis* and other mycobacteria. We screened 14 human gut-derived *Bacteroidetes* for arabinogalactan-degrading activities and identified four families of glycoside hydrolases with activity against the D-arabinan or D-galactan components of arabinogalactan. Using one of these isolates with exo-D-galactofuranosidase activity, we generated enriched D-arabinan and used it to identify a strain of *Dysgonomonas gadei* as a D-arabinan degrader. This enabled the discovery of endo- and exo-acting enzymes that cleave D-arabinan, including members of the DUF2961 family (GH172) and a family of glycoside hydrolases (DUF4185/GH183) that display endo-D-arabinofuranase activity and are conserved in mycobacteria and other microbes. Mycobacterial genomes encode two conserved endo-D-arabinanases with different preferences for the D-arabinan-containing cell wall components arabinogalactan and lipoarabinomannan, suggesting they are important for cell wall modification and/or degradation. The discovery of these enzymes will support future studies into the structure and function of the mycobacterial cell wall.

Growth and division of all bacteria is a carefully orchestrated process requiring the coordinated action of a host of enzymes. Acid-fast organisms such as *Mycobacterium tuberculosis* possess unusual cell wall glycans and lipids which require additional enzymatic machinery during this process. The core of the cell wall structure is conserved amongst mycobacteria and consists of three layers[1,2]. Like other bacteria, peptidoglycan forms the basal layer of the cell wall, though the precise architecture is unknown. At the other extremity of the wall are the mycolic acids which give the organisms their characteristic waxy appearance and are interspersed with a host of species-specific free lipids. Joining these two layers is a complex polysaccharide called

arabinogalactan (AG), which has a chemical composition unique to the Mycobacteriales and entirely distinct from the similarly named molecule found in plants that is composed of L-arabinofuranose (Fig. 1A)[3]. AG is comprised of two domains with a β-D-galactofuranose backbone decorated by large α-D-arabinofuranose branches[4]. The structure and biosynthesis of this molecule has undergone intense scrutiny, due in part to it being a target of the antimycobacterial drug ethambutol[5–8]. Ethambutol targets the arabinosyltransferase proteins in the cell envelope of mycobacteria that are responsible for biosynthesis of the polysaccharide[9]. Similarly, the biogenesis of mycolic acids and peptidoglycan are the subject of much research due to their biochemical

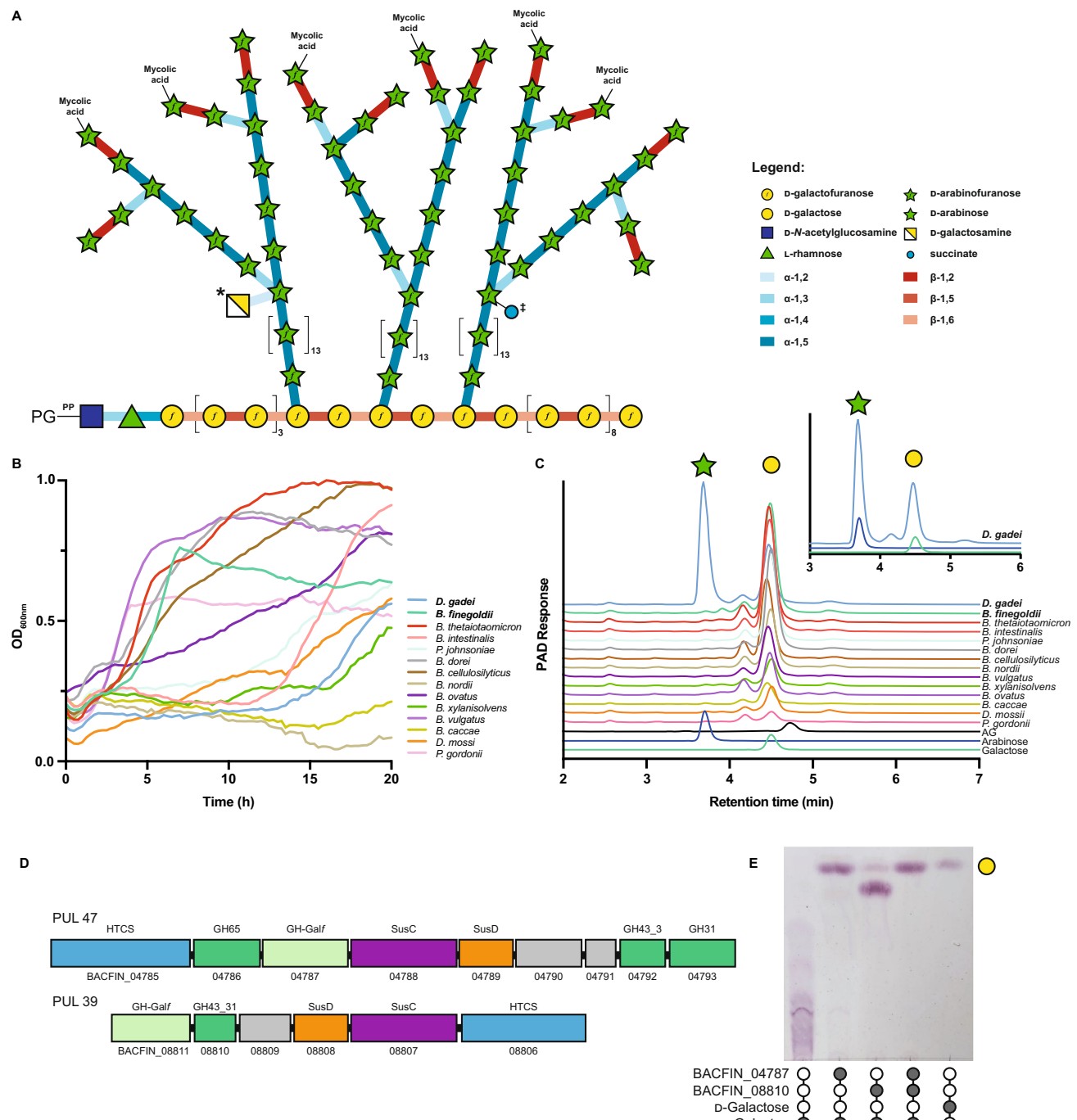

**Fig. 1 | Bacteroidetes growth on mycobacterial arabinogalactan. A** Schematic of the structure of mycobacterial mycolyl-arabinogalactan-peptidoglycan complex. Succinate (‡) and galactosamine (*) modifications are non-stoichiometric. **B** Growth of selected Bacteroidetes species on arabinogalactan as monitored by change in $OD_{600nm}$. **C** Ion chromatography with pulsed amperometric detection (IC-PAD) analysis of culture supernatants for bacteria grown on mycobacterial AG as a sole-carbon source. Production of arabinose (green star) and galactose (yellow circle) are indicated and compared to standards. Data from *D. gadei* is also shown as an inset. **D** Schematics of PUL47 and 39 from *B. finegoldii*, whose homologs in *B. cellulosilyticus* were identified by RNAseq as being upregulated during growth on AG. **E** TLC analysis of endpoint reaction products of *B. finegoldii* enzymes with *C. glutamicum* Δ*ubiA* galactan. Filled circles indicate the presence of a given reaction component. The major product appears to be galactobiose suggesting endo-activity. A detailed analysis of this enzyme is beyond the scope of this study. Source data are provided in the Source Data file.

complexity, essentiality, and their synthesis being the target of iso-niazid and β-lactams respectively[10,11].

The growth of mycobacteria likely requires not only the coordinated synthesis of the cell wall, but also its eventual degradation and turn-over to allow for cellular expansion, division and the insertion of cell envelope-spanning secretion systems. Similarly, the control of cell surface structures affords these bacteria the ability to modulate how they are sensed by the host. As such, hydrolases of the cell wall may play an important role at the mycobacterial cell surface. Hydrolases of peptidoglycan influence mycobacterial morphology and infection and it was recently demonstrated that at least some peptidoglycan by-products are recycled by the bacteria[12,13]. Similarly, trehalose-myco-lates, a dominant lipid component of the cell wall, undergo recycling[14], and a hydrolase involved in turning over mycolic acids (Rv3451) and a

recycling pathway for trehalose has been described[15]. Cleavage of the acid-fast cell wall is unlikely to be restricted to the Mycobacteriales. Organisms that predate on acid-fast bacteria such as phage or bacterial predators like the recently reported *Candidatus* Mycosynbacter amalyticus (hereafter *M. amalyticus*), are also likely to require D-arabinan degrading enzymes to penetrate the mycobacterial cell wall[16].

Only a single enzyme capable of degrading AG has been characterized to date. GlfH1 (Rv3096) is an exo-β-D-galactofuranolase from glycoside hydrolase (GH) family GH5_13 that cleaves the galactan backbone of mycobacterial AG at both β−1,5 and β−1,6 linked residues[17]. The precise role of this enzyme in mycobacterial biology remains unclear but its activity is suggestive of it being part of a remodeling pathway for this cell wall component. Other enzymes with activity on simple D-galactofuranosides (D-Gal*f*) have been identified through large-scale screening of orphan GH sequences[18]. However, no enzymes have been characterized with exo- or endo-D-arabinofuranase activity (enzymes that cut within the D-arabinan polymers) against AG, although this activity was described in protein extracts from soil bacteria dating back to the 1970s[19,20]. Endo-D-arabinanase activity was also described in extracts of *Mycobacterium smegmatis* and was suggested to increase upon treatment of cells with ethambutol, which blocks D-arabinan biosynthesis[21,22]. During preparation of this manuscript, an exo-acting difructose-dianhydride I synthase/hydrolase was discovered that was also active on *p*NP-α-D-arabinofuranoside[23]. Whether this enzyme acts on mycobacterial AG is unknown.

The human gut microbiota is responsible for the degradation of dietary plant polysaccharides, host and microbial glycans. Dominated by the Bacteroidetes, this grouping of organisms is collectively amongst the richest known organisms in the diversity of complex carbohydrate degrading enzymes[24]. Carbohydrate utilization by the Bacteroidetes is typically mediated by genes, which are organized into polysaccharide utilization loci (PULs), that can be induced upon exposure of the bacterium to a given carbohydrate[25–28]. The abundance and diversity of carbohydrate-degrading enzymes in the Bacteroidetes provides a rich opportunity for enzyme discovery.

In this study we have mined the glycolytic capacity of the human gut microbiota and discovered a collection of enzymes able to completely degrade mycobacterial arabinogalactan. We report the discovery of glycoside hydrolases active on the mycobacterial cell wall. We also identify exo-D-arabinofuranosidases from the DUF2961 family (GH172). Furthermore, we demonstrate that D-arabinan degradation is wide-spread amongst Mycobacteriales and mycobacteria, but is also present in phages, other bacteria, and microbial eukaryotes. Our data point to a key role for these enzymes in mycobacterial biology and can enable sophisticated analysis of mycobacterial cell wall components.

## Results
### Select human gut Bacteroidetes can utilize mycobacterial AG as a carbon source
While endo-D-arabinofuranase activity was first described in soil bacteria more than 50 years ago and has been known in mycobacteria for at least 30 years[19,20], the enzymes responsible for this activity have escaped identification. We reasoned that these enzymes might be highly regulated, unstable, or poorly soluble in mycobacteria making purification-based approaches unfeasible. Moreover, if they belong to novel enzyme class(es), bioinformatics approaches would fail. Instead, we isolated arabinogalactan from *M. smegmatis* mc²155 and used it as sole-carbon source for the growth of a panel of 14 Bacteroidetes species. Of these, 12 strains were able to grow on this material (Fig. 1B). Ion chromatography with pulsed amperometric detection (IC-PAD) analysis of selected culture supernatants (Fig. 1C) demonstrated the production of free galactose in most cultures, and arabinose in one; *Dysgonomonas gadei* (Fig. 1C, inset). Together these data indicate that members of the gut microbiota produce enzymes that can depolymerize mycobacterial D-arabinan and D-galactan.

### Identification of exo- and endo-galactofuranosidases that degrade galactan
The presence of both galactose and arabinose in *D. gadei* culture supernatants complicated the identification of PULs specific for either galactan or arabinan. Therefore, we developed a method for production of pure D-arabinan by exploiting D-galactan specific PULs. Based on the analysis of culture supernatants, *Bacteroides finegoldii* and *Bacteroides cellulosilyticus* appeared to degrade galactan, but not D-arabinan. RNAseq analysis of *B. cellulosilyticus* revealed the upregulation of PUL35 and PUL36, containing multiple predicted GHs (Figure S1). While we could not heterologously express the *B. cellulosilyticus* enzymes, the homologs of these enzymes from *B. finegoldii* DSM17565 (PULDB ID: PUL39 and PUL47, Fig. 1D) could be expressed and purified, and when tested on galactan (Fig. 1E) demonstrated exo- and endo- activities. To determine their galactan degradative capacity, we purified galactan, comprised of alternating β−1,5- and β−1,6-galactofuranose residues from a strain of *Corynebacterium glutamicum* (ΔubiA) that lacks D-arabinan[29]. As shown in Fig. 1E and Figure S1, combination of the two *B. finegoldii* enzymes comprising a GH43_31 (BACFIN_08810) and a previously unidentified exo-Gal*f* (BACFIN_04787) family completely hydrolysed this galactan substrate.

### Identification of a D-arabinan-degradation PUL
To generate D-arabinan we digested mycobacterial AG with the two *B. finegoldii* galactofuranosidases (BACFIN_08810 and BACFIN_04787). This resulted in an enriched D-arabinan fraction with approximately 70% reduction in galactan as determined by acid hydrolysis (Figure S1). The resulting D-arabinan was used as a sole carbon source for *D. gadei* (which was previously shown to produce both D-galactose and D-arabinose from AG). Proteomics of these bacteria at mid-log phase during growth on enriched D-arabinan identified a predicted fucose isomerase as the most abundant carbohydrate-active enzyme (Supplementary Data 1). This protein maps to PUL42 in the *D. gadei* genome (Fig. 2A), and an additional nine of the proteins derived from this PUL were in the top 200 most abundant proteins in the total proteome, including several that lacked annotation (Figure S2). Reasoning that mycobacteria may harbor homologs of *D. gadei* arabinanases to process arabinan, we prioritized the DUF2961 and DUF4185 superfamily enzymes encoded in PUL42 as they possessed homology to predicted mycobacterial proteins of unknown function within the same superfamilies.

### The DUF2961 superfamily (GH172) includes D-arabinofuranosidases
At the outset of this study, no member of the DUF2961 family had been characterized. Therefore, we cloned, expressed, and purified all three DUF2961 family members found in PUL42 in *D. gadei*. Upon incubation with purified AG, we observed D-arabinofuranosidase activity for HMPREF9455_02467, HMPREF9455_02471 and HMPREF9455_02479 (hereafter Dg_GH172a, Dg_GH172b and Dg_GH172c, respectively) (Figure S3). To probe the activity of these enzymes, we synthesized the chromogenic substrates *p*-nitrophenyl α-D-arabinofuranoside (pNP-α-D-Ara*f*) and β-D-arabinofuranoside (pNP-β-D-Ara*f*). Dg_GH172a and Dg_GH172c were active against pNP-α-D-Ara*f*, and no activity was observed using pNP-β-D-Ara*f* (Table 1). Although substrate limitations prevented accurate determination of $V_{max}$ we could use pNP-α-D-Ara*f* to determine $k_{cat}/K_M$ for Dg_GH172c (Figure S4 and Table 2). These data demonstrate that gut bacteria can use DUF2961 enzymes to generate D-arabinose from mycobacterial arabinogalactan.

### DUF2961 superfamily (GH172) enzymes are present in acid-fast bacteria and their predators
Homologues of the DUF2961 encoding genes are present in the genomes of organisms from the Actinomycetota phylum including

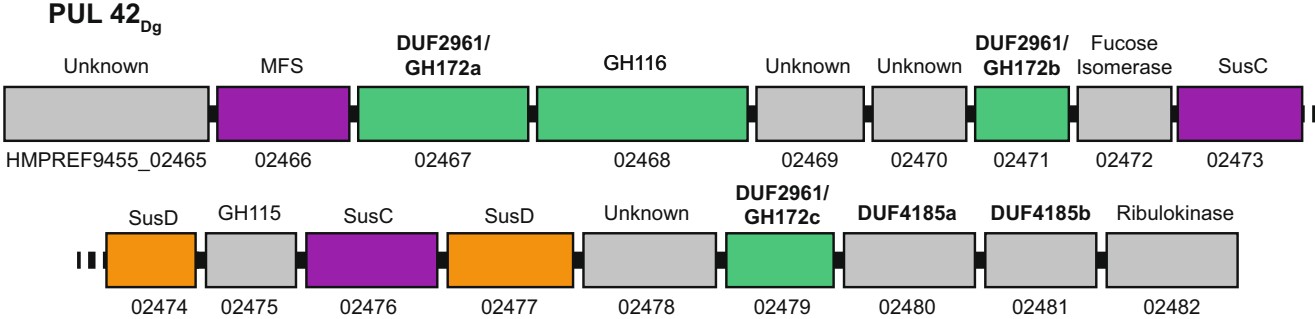

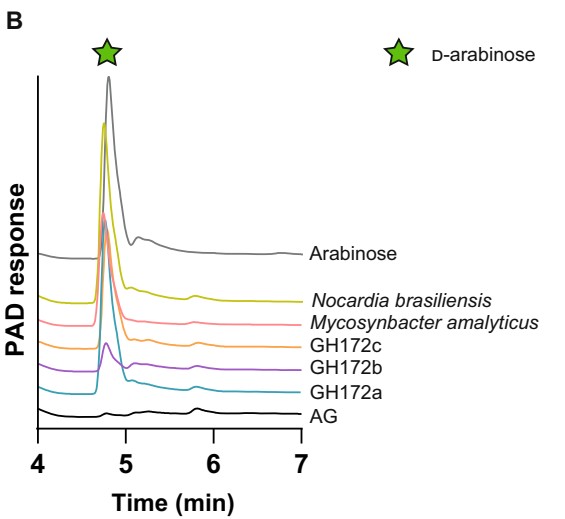

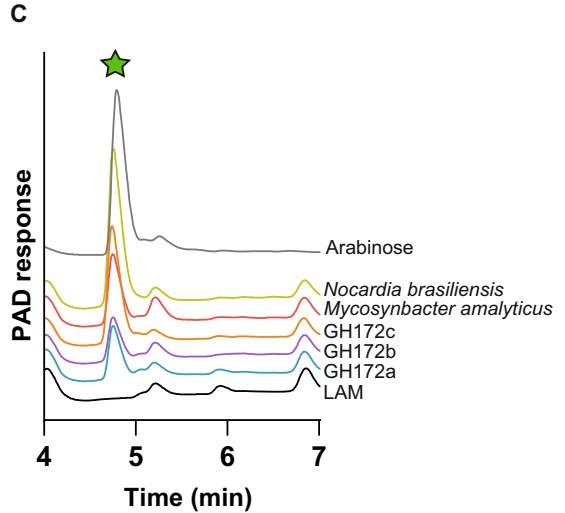

**Fig. 2 | *D. gadei* DUF2961 genes encode GH172 exo-ᴅ-arabinofuranosidases with orthologs in diverse bacteria. A** Schematic of PUL42 as identified by proteomic analysis of *D. gadei* grown on ᴅ-arabinan. **B** DUF2961 enzymes (1 μM) vs 2 mg ml⁻¹ AG and (**C**) LAM. Samples were analyzed by ion chromatography with pulsed amperometric detection (IC-PAD) on a Dionex ICS-6000 with CarboPac Pa300 column, 100 mM NaOH, 20 min isocratic elution followed by a 0-60% 500 mM sodium acetate gradient over 60 minutes. Green star = ᴅ-arabinose. Source data are provided in the Source Data file.

pathogens such as *Mycobacterium avium* subsp. *paratuberculosis* (MAP_0339c) and *Nocardia brasiliensis* (O3I_017420; Noc$_{GH172}$). To validate the activity of these enzymes, we attempted to produce each recombinantly. Noc$_{GH172}$ was soluble; however, despite repeated attempts, we were unable to produce soluble MAP_0339c. Noc$_{GH172}$ had ᴅ-arabinofuranosidase activity on both pNP-α-ᴅ-Ara*f* and purified arabinogalactan (Table 2, Figure S3). We also identified a DUF2961 homolog within the genome of the parasitic bacterium *M. amalyticus* (GII36_05205; Myc$_{GH172}$), which possessed similar activity (Fig. 2 and S3A)[16]. Together these data indicate that exo-ᴅ-arabinofuranosidase activity is a feature of the DUF2961 superfamily. These enzymes are encoded by both the Mycobacteriales and their predators.

## Table 1 | Activity of GH172 enzymes against synthetic and bacterial substrates

| Protein | pNP-α-ᴅ-Ara*f* | AG | LAM | Pili |
|---|---|---|---|---|
| Dg$_{GH172a}$ | ✓ | ✓ | ✓ | ✓ |
| Dg$_{GH172b}$ | weak | ✓ | ✓ | weak |
| Dg$_{GH172c}$ | ✓ | ✓ | ✓ | ✓ |
| Noc$_{GH172}$ | ✓ | ✓ | ✓ | ✓ |
| Myc$_{GH172}$ | ✓ | ✓ | ✓ | weak |

## GH172 enzymes degrade α−1,5-ᴅ-arabinofuranoside linkages within ᴅ-arabinan

A major limitation of studying these enzymes is access to substrates with defined glycosidic linkages. To circumvent this, we took advantage of the discovery that *Pseudomonas aeruginosa* PA7 decorates its pili with short α−1,5-ᴅ-Ara*f* containing oligosaccharides which can be readily accessed through established protocols[30]. To understand the substrate specificity of these enzymes against other microbial ᴅ-arabinan substrates, we assayed our DUF2961 GH172 enzymes (Dg$_{GH172a}$, Dg$_{GH172b}$, Dg$_{GH172c}$, Noc$_{GH172}$, and Myc$_{GH172}$) against mycobacterial LAM and ᴅ-Ara*f* containing pilin oligosaccharides (Table 1). The ᴅ-arabinan branches of lipoarabinomannan (LAM) are structurally similar to those in AG. All enzymes could digest LAM from *M. smegmatis* to produce arabinose, although Dg$_{GH172b}$ and Myc$_{GH172}$ displayed lower activity than the others (Fig. 2C and Figure S3B).

Similar to our results with LAM, all of the GH172 enzymes except for Dg$_{GH172b}$, could cleave the ᴅ-arabinofuranose oligosaccharides from digested pili (Figure S3C). To rule out ᴅ-galactofuranosidase activity, we repeated these experiments using purified ᴅ-galactan (Figure S3D) and observed no activity. The combination of these experiments reveals that GH172 enzymes are specific for α-ᴅ-arabinofuranoside linkages, are active on ᴅ-arabinan, and can cleave α−1,5-ᴅ-arabinofuranoside linkages.

**Table 2 | $k_{cat}/K_M$ of GH172 enzymes for pNP-α-D-Araf and AG**

| Protein | pNP-D-Araf |
|---|---|
| Dg$_{GH172a}$ | n/a |
| Dg$_{GH172c}$ | $2.9 \times 10^4 \pm 4 \times 10^2 \, min^{-1} \, M^{-1}$ |
| Noc$_{GH172}$ | $1.2 \times 10^5 \pm 2 \times 10^3 \, min^{-1} \, M^{-1}$ |

**Table 3 | SEC-LS oligomerization state of GH172 enzymes**

| Protein | Observed MW (Da) | Oligomerization state | Theoretical MW of oligomer (Da) |
|---|---|---|---|
| Dg$_{GH172c}$ | 260,829 | hexamer | 266,952 |
| Noc$_{GH172}$ | 123,934 | trimer | 120,507 |
| Dg$_{GH172a}$ | 515,568 | dodecamer | 519,438 |
| Dg$_{GH172b}$ | 148,350 | dimer | 146,512 |
| Myc$_{GH172}$ | 192,623 | trimer | 202,115 |

## Family GH172 proteins can adopt diverse oligomeric states

To better understand the structure-function relationship of this protein family, we examined the multimeric state of several of our candidates of interest. Size-exclusion chromatography with laser scattering (SEC-LS) analysis of the GH172 family enzymes allowed assessment of molecular weight and assignment of oligomerization states (Table 3 and Figure S5). A wide range of oligomeric states were uncovered: Dg$_{GH172c}$ was assigned as a hexamer, Dg$_{GH172a}$ a dodecamer, Noc$_{GH172}$ and Myc$_{GH172}$ as trimers, and Dg$_{GH172b}$ as a dimer. This solution data complements recent reports of hexameric assemblies in protein crystals of two GH172 members: difructose dianhydride I synthase/hydrolase from *Bifidobacterium dentium* (PDB: 7V1V) and BACUNI_00161 from *Bacteroides uniformis* (PDB: 4KQ7)[23].

Interestingly, Dg$_{GH172b}$ is predicted to have 1.5 DUF2961 domains, and so dimerization of Dg$_{GH172b}$ is predicted to provide three DUF2961 domains in the final assembly. SEC-LS analysis is supportive of a quaternary structure for each of the GH172 proteins exhibiting multiple DUF2961 domains: Dg$_{GH172b}$, Myc$_{GH172}$ and Noc$_{GH172}$ each contain three DUF2961 domains, while Dg$_{GH172c}$ contains six and Dg$_{GH172a}$ contains twelve. Diversity in quaternary structure may lead to differences in activity or substrate specificity.

## Dg$_{GH172c}$ catalysis is driven by conserved glutamate residues

To gain insight into the functional roles of the residues in the active site of Dg$_{GH172c}$, the proposed catalytic carboxylate residues were mutagenized to alanine, generating variants E233A, E254A and D225A. Dg$_{GH172c}$-E233A and Dg$_{GH172c}$-D225A had no detectable activity and there was a greater than $10^5$-fold reduction in activity for Dg$_{GH172c}$-E254A. Our data support the assignment of the conserved E254 and E233 as catalytic residues (acid/base or nucleophile) (Figure S6) and highlight an important role for the adjacent D255 residue. None of the variants displayed a change in oligomerization state suggesting they do not possess structural roles (Figure S7).

## DUF4185 is widely distributed throughout bacterial species

In Bacteroidetes, the degradation of a target polysaccharide is typically a multi-step process whereby oligosaccharides are generated and subsequently cleaved into their monosaccharide constituents, encoded within co-transcribed operons. We reasoned that the generation of D-arabinofuranose oligosaccharides was likely achieved by proteins that have no annotated function and so focused our attention on the DUF4185 proteins. To investigate the conservation of these genes across different organisms, we constructed a phylogeny of DUF4185 proteins. This revealed that they are common within actinobacteria, especially amongst the Actinomycetota (Fig. 3 and Figure S8), in *Bacteroides* species, *Myxococcus*, the lysis cassette of some actinobacteriophage (Figure S9) and the predatory bacterium *M. amalyticus*.

## DUF4185 comprises a GH family with endo-D-arabinanase activity

Initially, we cloned, expressed, and purified the DUF4185 homologs (HMPREF9455_02480 and HMPREF9455_02481) from *D. gadei* (herein, referred to as Dg$_{GH4185a}$ and Dg$_{GH4185b}$, respectively). When incubated with mycobacterial AG and then analyzed by HPAEC, these enzymes produced a banding pattern characteristic of an endo-acting GH (Figure S10A), consistent with endo-arabinanase activity. To assess the breadth of activity for DUF4185 enzymes we selected additional candidate enzymes from each of the major lobes of the global phylogeny (Fig. 3). Recombinant proteins were produced from several bacterial lineages and a phage capable of infecting Gram-positive bacteria *Gordonia* of the order Mycobacteriales. These included *Myxococcus xanthus* (Myxo$_{GH4185}$), *M. amalyticus*, and *Gordonia* Phage GMA6 (Phage$_{GH4185}$). Where the DUF4185 domain sat within a larger gene containing several other large domains, we produced truncated variants containing only the DUF4185 domain due to low solubility of the multidomain proteins. All DUF4185 constructs except that from *M. amalyticus* yielded soluble protein. As shown in Fig. 4 and Figure S10, when incubated with mycobacterial AG all these enzymes possessed endo-D-arabinanase activity with varying product profiles suggesting differences in enzyme specificity. Some of the enzymes were also active against the linear α−1,5-D-arabinofuranose oligosaccharides derived from *P. aeruginosa* PA7 pili, consistent with activity against the major linkage of mycobacterial D-arabinan (Figure S10). The discovery of endo-D-arabinanase activity in proteins from organisms outside of the Actinomycetota is an interesting observation, however its presence in bacterial predators is consistent with the essentiality of this polymer for mycobacterial viability. This sugar motif has been reported in a small number of LPS structures, providing a possible explanation for the presence of this enzymatic activity in the gut microbiota[31,32]. Furthermore, some corynebacterial species are found as commensals of the human oral and gut microbiome, which may cross-feed these bacteria through shedding of cell wall structures[33].

## Mycobacterial DUF4185s are endo-D-arabinofuranases

Given the importance of D-arabinan to mycobacterial viability and immunology we next sought to understand the biochemical function of DUF4185 proteins from representative mycobacteria. As shown in Fig. 3 and Figure S8, mycobacteria produce at least two DUF4185s that fall into distinct phylogenetic groupings. In *M. tuberculosis* these are Rv1754c and Rv3707c. Beyond these two conserved DUF4185 genes, some species have additional DUF4185 members. For example, many *Mycobacterium abscessus* strains encode at least three distinct members whilst *M. smegmatis* mc²155 encodes five (MSMEG_4352, 4360, 4365, 2107 and 6255). Based on sequence analysis, MSMEG_2107 and MSMEG_6255 are homologs of Rv1754c and Rv3707c, respectively, while the remainder show greater diversity (Figure S8). A distant homolog of Rv3707c from *Mycobacterium abscessus*, Ga0069448_1118 (hereafter referred to as *Mab$_{4185}$*), was readily produced in soluble form and in good yield[34]. Despite low sequence identity (33.5%) (Figure S11) to the *D. gadei* enzymes, the HPAEC profiles of mycobacterial AG digested by Mab$_{4185}$ indicate that it also possesses endo-D-arabinofuranase activity (Fig. 4).

Encouraged by this success with *Mab$_{4185}$* but recognizing it has limited sequence identity with either of the *M. tuberculosis* proteins (17% and 16.4% identical to Rv3707c and Rv1754c respectively over the entire protein length), we sought to study Rv3707c and Rv1754. However, despite our best efforts we were unable to produce usable amounts of soluble Rv3707c and Rv1754. A previous report highlighted that Rv3707c is secreted, but it lacks a discernible signal peptide[35]. We reasoned that the instability of Rv3707c may be due to incorrect

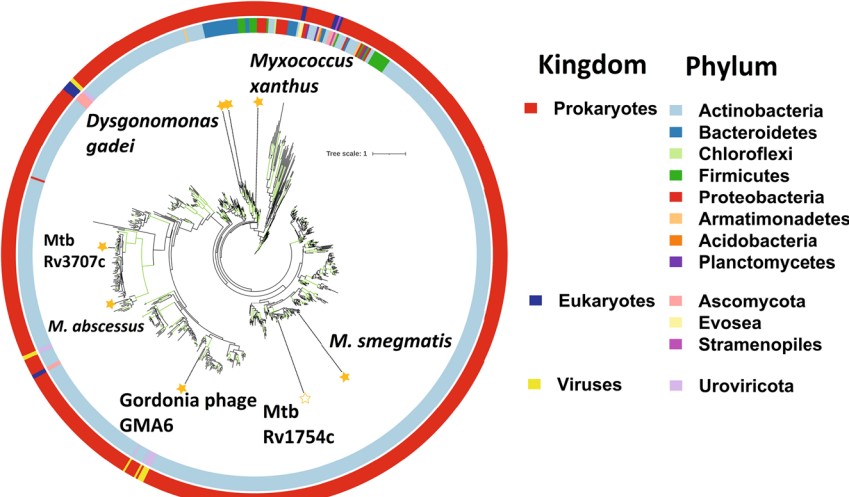

**Fig. 3 | Phylogeny of the DUF4185 enzyme family.** Unrooted ML phylogeny (LG model with empirical base frequencies, invariable sites and the discrete gamma model) of DUF4185 family sequences. Branches with greater than 75% bootstrap support (100 replicates) are highlighted in green. Units for tree scale are inferred substitutions per amino acid residue. Colored rings indicate phylum (inner) and kingdom (outer) taxonomy information for sequences. Stars highlight sequences of interest and are filled for proteins that have been experimentally characterized in this work.

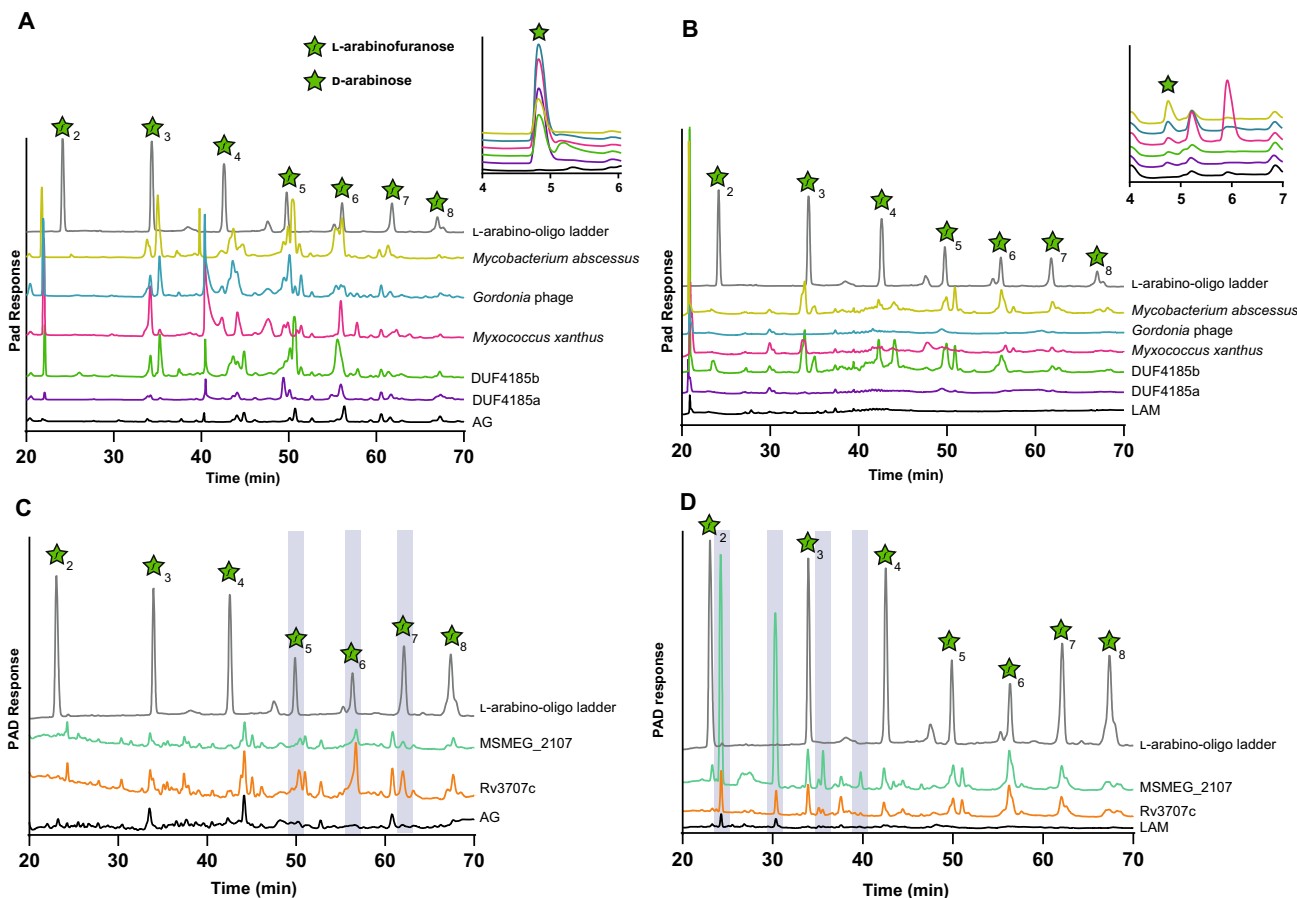

**Fig. 4 | DUF4185 proteins are endo-ᴅ-arabinofuranases that cleave myco-bacterial AG and LAM.** DUF4185 enzymes were incubated with 2 mg•ml⁻¹ AG (**A** and **C**) or 2 mg · ml⁻¹ LAM (**B** and **D**) for 16 hours as described in Materials and Methods. Samples were analyzed by ion chromatography with pulsed ampero-metric detection (IC-PAD) on a Dionex ICS-6000w with CarboPac Pa300 column, 100 mM NaOH 20 min isocratic elution followed by a 0-60% 500 mM sodium acetate gradient over 60 minutes. A ladder of α-1,5-ʟ-arabino-oligosaccharides (25 μM) derived from plant arabinan was used as a standard. In panel C the chro-matogram of this ladder has been scaled on the *y*-axis by a factor of 0.2 for clarity of presentation. Source data are provided in the Source Data file.

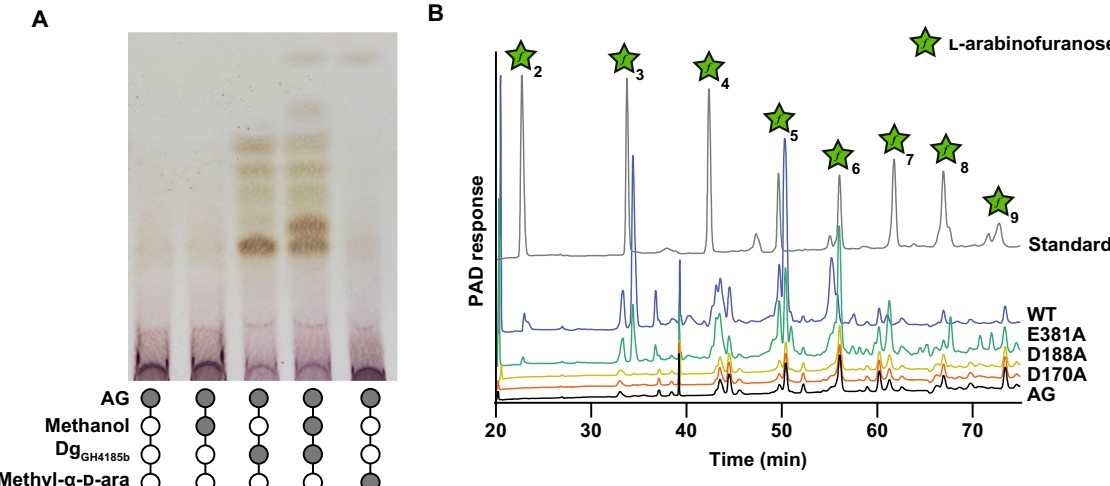

**Fig. 5 | Catalytic analysis of DUF4185 endo-ᴅ-arabinofuranases. A** Methanolysis of AG catalyzed by $Dg_{GH4185b}$. The production of methyl arabinosides in the presence of methanol indicates a retaining mechanism. **B** $Dg_{GH4185b}$-WT, $Dg_{GH4185b}$-D149A and $Dg_{GH4185b}$-D167A were incubated at 1 μM with 2 mg ml⁻¹ AG for 16 hours and analyzed by ion chromatography with pulsed amperometric detection (IC-PAD). No activity was observed for the D188A or D170A mutants. Source data are provided in the Source Data file.

annotation of the start site in the *M. tuberculosis* H37Rv genome, given that mycobacteria frequently use alternative start codons in addition to the canonical ATG. We re-evaluated the genomic context of the protein and identified several possible in-frame N-terminal extensions of the gene (Figure S12). We compared these potential N-terminal extensions to alignments of DUF4185 proteins that possessed a second, N-terminal domain to identify sequence motifs that are conserved; in many cases a proline-rich region was observed, which is also found in the potential N-terminal extensions of Rv3707c (Figure S12). Furthermore, by extending the N-terminus of the protein, a putative signal peptide can be predicted by SignalP 6.0 (Figure S12)[36]. While the precise start-site is uncertain, the likely cleavage point for the signal peptide could be confidently assigned. Therefore, we designed an expression construct with the putative signal peptide removed, but the remainder of the proline-rich N-terminus intact. This produced a reasonably stable and soluble Rv3707c protein in good yield that digested AG to give products consistent with a hepta- or hexa-saccharide and was also active on LAM (Fig. 4C/D). While we were unable to produce soluble Rv1754c, we successfully produced the *M. smegmatis* homolog (MSMEG_2107), which demonstrated activity against LAM, but not AG (Fig. 4C/D and Figure S10).

### DUF4185 sub-clades have distinct substrate specificities

To further probe the substrate specificity of these enzymes and elucidate the function of MSMEG_2107, we incubated each of the DUF4185 enzymes ($Dg_{GH4185a}$, $Dg_{GH4185b}$, $Mab_{GH4185}$, $Phage_{GH4185}$, $Myxo_{GH4185}$, MSMEG_2107, and Rv3707c) with: AG or LAM (Figure S10A, B); purified ᴅ-galactan (Figure S10C); and pilin oligosaccharides from *P. aeruginosa* PA7 (Figure S10D). The majority of the enzymes displayed higher activity against AG than LAM. Conversely, MSMEG_2107 was unique in displaying a preference for LAM with no detectable activity when incubated with AG under similar conditions. To confirm that the enzymes only degraded ᴅ-arabinan and not ᴅ-galactan we tested their activity with ᴅ-galactan but did not observe any product formation (Figure S10C). Given the presence of ᴅ-arabinan motifs in both AG and LAM, we conclude the DUF4185 family are endo-ᴅ-arabinofuranases. We note that only a subset of the DUF4185 enzymes were active against pili from *P. aeruginosa* PA7. As these oligosaccharides are relatively short and linear, this suggests specific enzyme subsite occupancy for activity, which leads to production of arabinose for some enzymes; by contrast arabinose production was not observed upon digestion of branched, polymeric AG.

We next assessed whether DUF4185 endo-ᴅ-arabinofuranases can cleave AG in the context of an intact cell wall. We utilized metabolic arabinogalactan labelling with the azide-modified lipid-linked Araf donor 5-AzFPA and then fluorescently labelled intact bacteria with DBCO-conjugated AF647 using click chemistry (Figure S13)[37–39]. Both $Mab_{GH4185}$ and Rv3707c released fluorescently labelled material from cell walls, with greater activity produced by the latter enzyme. This contrasted with what was observed by IC-PAD using isolated AG. Likewise, treatment with $GH172_{Noc}$ led to release of fluorescent products, supporting the conclusion that both groups of enzymes can cleave AG in the context of the intact mycobacterial cell wall.

### DUF4185 family are anomer-retaining enzymes

Glycoside hydrolases can hydrolyze the anomeric linkage through either reversion or retention. Inclusion of a simple alcohol, such as methanol, in an enzymatic digest can be used to identify retaining enzymes, as they may afford methylated glycosides[40]. Addition of methanol to AG digests by $Dg_{GH4185b}$ produced methyl arabinoside, thereby demonstrating a retaining mechanism (Fig. 5A). Three conserved carboxylate residues are predicted from sequence alignments (Figure S11). Using site-directed mutagenesis, we varied these carboxylate residues in $Dg_{GH4185b}$ to generate $Dg_{GH4185b}$-D170A and $Dg_{GH4185b}$-D188A and $Dg_{GH4185b}$-E381A derivatives. Activity was broadly retained for glutamate substitution, but no activity was observed for the two aspartate mutants (Fig. 5B). These data support a retaining mechanism for the DUF4185 family of enzymes and assignment of D170/D39 in $Dg_{GH4185b}$ (and D188/D56 in Rv3707c) as the catalytic residues corresponding to acid/base and nucleophile.

## Discussion

Endo-ᴅ-arabinanase activity was first reported more than 50 years ago, but despite the widespread availability of mycobacterial genomic tools and -omics technologies, these enzymes have escaped identification[22,41]. We have mined members of the the human gut microbiome and leveraged evolutionary conservation to identify these enzymes in mycobacteria, along with exo-ᴅ-arabinofuranosidases and exo-ᴅ-galactofuranosidases (Fig. 6). We reasoned that the abundance of Mycobacteriales in environmental niches in addition to the availability of ᴅ-arabinofuranose polymers in organisms such as *P. aeruginosa* PA7 and corynebacteria meant that the capacity to degrade this carbohydrate was likely to be encoded in the human gut microbiota[30,42], allowing

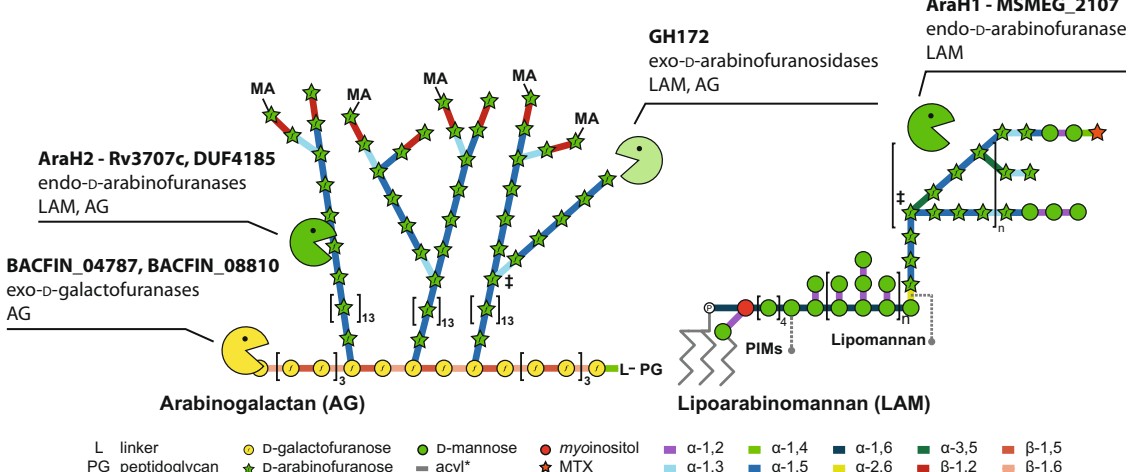

**Fig. 6 | Mycobacterial arabinogalactan-degrading enzymes discovered in this study and their substrates.** Each enzyme or enzyme family is listed with its identified function and the mycobacterial cell wall component it acts upon. MA mycolic acids, L linker unit, PG peptidoglycan, PIMs phosphatidylinostol mannosides.

identification by metabolic methods with arabinogalactan as sole carbon source.

Initially, we identified D-galactofuranase activity in a gut microbe organism, whose presence may reflect the widespread occurrence of D-Gal*f* in the LPS of some gram-negative bacteria of the human gut microflora[43]. One of the enzymes we identified, BACFIN_04787, is the founding member the glycoside hydrolase family GH182. The galactan degradation pattern exhibited by BACFIN_04787 is consistent with an exo-acting, non-specific enzyme and the product pattern of BAC-FIN_08810 is consistent with an exo-acting enzyme that can cleave either the β-1,5 or β-1,6 linkage within D-galactan. Further characterization of these enzymes will enable detailed analysis of mycobacterial galactan, whose chain length was recently shown to be important in the biology of these organisms[4].

Identifying organisms that can grow on D-arabinan is complicated by the presence of D-galactan in AG. To overcome this, we used the D-galactofuranase described here to generate enriched D-arabinan, which we used to identify *D. gadei* as a potent D-arabinan degrader. The PULs associated with this ability contained uncharacterized DUFs as well as DUF2961 genes corresponding to family GH172 which we have shown are exo-D-arabinofuranosidases. During the preparation of this manuscript, members of family GH172 were independently shown to exhibit exo-D-arabinofuranosidase activity by Kashima et al., consistent with our data[23]. SEC-LS analysis of a range of family GH172 proteins reveals diverse quaternary structures of multimers up to dodecamers consistent with the hexamer arrangement of the enzyme described by Kashima et al. How this structural diversity is connected to function is as-of-yet unexplored. It has been suggested that family GH172 proteins and phage capsid proteins may be ancestrally related, raising the possibility for diversification of function within this family[44].

The presence of genes encoding DUF4185 and GH172 enzymes in *M. amalyticus* suggests that they contribute to its epibiotic lifestyle on host *Gordonia* spp. through feeding on arabinogalactan as a carbon source. The Myc_DUF4185 enzyme is predicted to be a surface-located lipoprotein, while Myc_GH172 is predicted to not be secreted. This localization is consistent with a model whereby *M. amalyticus* releases oligosaccharides from the surface of *Gordonia*, internalizes them, and then cleaves them to monosaccharides in the cytoplasm. An alternative explanation is that the DUF4185 and GH172 enzymes locally remodel the cell wall of *Gordonia*, enabling access to cellular contents. In contrast, the biological role of GH172 enzymes in *Bacteroides* species remains unclear. While some may be involved in α-fructan degradation, those associated with D-arabinan PULs are more likely targeted at

either glycans derived from Actinomycetota or organisms such as *P. aeruginosa* PA7.

The identification of the DUF4185 family of enzymes connects a GH family to the endo-D-arabinofuranase activity reported more than 50 years ago. In 1972 Kotani and colleagues reported the isolation of "mixed D-arabinanase activity" in an extract from an unnamed Gram-positive soil microbe, referred to as the "M-2" fraction[19]. This extract possessed endo-activity and released a wide range of products from mycobacterial cell walls, although the specific protein responsible was not identified. Since then, similar impure enzyme cocktails have contributed to numerous studies on AG and LAM[41]. We have now identified broadly conserved endo-D-arabinofuranases from the Mycobacteriales, which we propose to name AraH1 (Rv1754c) and AraH2 (Rv3707c). These enzymes are the founding members of the GH183 family. The availability of well-characterized enzymes with defined activities should support more detailed studies of mycobacterial cell wall polymers.

A review of functional screens of knockout libraries of *M. tuberculosis* also highlights an important role for AraH2 in pathogenesis. AraH2 was identified in a screen for proteins with non-canonical signal sequences[35], likely because of mis-annotation of its signal peptide as demonstrated by our work. In that study, Perkowski and colleagues reported that a transposon mutant in AraH2 was severely defective for replication in macrophages. A separate study identified AraH2 as important for control of phagosome acidification, where it was the second most enriched mutant in an acidified phagosome screen[45]. A homologue of AraH2, PEG_1752, was shown to be upregulated in the phylogenetically related *Mycobacterium llatzerense* upon infection of the amoeba *Acanthamoeba castellanii*[46]. These data point to a role for AraH2 in phagosome survival, and by extension of D-arabinan remodeling in mycobacterial pathogenesis.

The role of AraH1 remains more elusive. The gene is broadly conserved amongst mycobacteria, and our biochemical data suggests this enzyme class is active against LAM, but not AG. This genomic locus is a frequent site of IS6110 element insertion[47], and interruption of this gene could cause LAM structural variability amongst circulating strains of *M. tuberculosis*. It is also possible that AraH1 and AraH2 have partially overlapping substrate specificity and the former can partially compensate for the loss of the latter. As well, these proteins may have specific roles under narrowly defined conditions. Analogously, peptidoglycan-lytic enzymes with seemingly redundant reaction specificities are encoded in most bacteria and lack notable phenotypes for their loss under most growth conditions.

However, screens at a range of pH values have identified specific functions for these enzymes[48].

To conclude, we have unearthed a new enzymatic toolkit for the degradation of mycobacterial arabinogalactan that should find utility in the study of the structure and function of this important polymer (Fig. 6). The functional annotation of these genes will support future investigations of the role of D-arabinan structural modulation in mycobacterial biology, mycobacteriophage infection and interbacterial predation.

## Methods

### Bacterial strains and growth conditions

Bacteroidetes sp. were grown on a 2x defined minimal media (Table S1) under anaerobic conditions at 37 °C over 24 hours to assay growth on various carbon sources, including arabinogalactan. Strains used were *Bacteroides caccae* ATCC 43185, *B. cellulosilyticus* DSM 14838, *B. dorei* DSM 17855, *B. finegoldii* DSM 17565, *B. intestinalis* DSM 17393, *B. nordii* CL02T12C05, *B. ovatus* ATCC 8483, *B. thetaiotaomicron* VPI-5482, *B. vulgatus* ATCC 8482, *B. xylanisolvens* XB1A, *Dysgonomonas gadei* ATCC BAA-286, *D. mossii* DSM 22836, *Parabacteroides gordonii* DSM 23371, *P. johnsonii* DSM 18315. *Escherichia coli* and *Pseudomonas aeruginosa* PA7 ATCC 15692 strains were grown in lysogeny broth at 37 °C (unless otherwise specified). *Mycobacterium smegmatis* mc²155 ATCC 19420 was grown in Tryptic soy Broth at 37 °C with agitation.

### RNA sequencing

*B. cellulosilyticus* was cultured in defined media (Table S2) containing 5 mg ml⁻¹ AG or glucose, in triplicate 5 ml cultures. Cells were harvested at mid-log phase and stored in RNA protect (Qiagen). RNA was purified with the RNAeasy Kit. Prior to library preparation, rRNA was depleted using the Pan-Prokaryote riboPOOLs kit (siTOOLs Biotech). In brief, 1 μg of total RNA was incubated for 10 min at 68 °C and 30 min at 37 °C with 100 pmol of rRNA-specific biotinylated DNA probes in 2.5 mM Tris-HCl pH 7.5, 0.25 mM EDTA, and 500 mM NaCl. DNA-rRNA hybrids were depleted from total RNA by two consecutive 15 min incubations with 0.45 mg streptavidin-coated magnetic Dynabeads MyOne C1 (ThermoFisher Scientific) in 2.5 mM Tris-HCl pH 7.5, 0.25 mM EDTA, and 1 M NaCl at 37 °C. The rRNA-depleted RNA samples were purified using the Zymo RNA Clean & Concentrator kit combined with DNase treatment on a solid support (Zymo Research).

cDNA libraries were prepared using the NEBNext Multiplex Small RNA Library Prep kit for Illumina (NEB) in accordance with the manufacturers' instructions.

Library preparation and sequencing took place at the Earlham Institute, and were processed by Newcastle University Bioinformatics Support Unit. Briefly, raw sequencing reads were checked using Fast QC, reads were mapped to *Bacteroides cellulosilyticus* DSM 14838 (GCA_000158035) downloaded from Ensembl (assembly ID ASM15803v1). Reads were quantified against genes contained in the Ensembl annotation using featureCounts from the Rsubread package[49]. Counts were normalized by Trimmed Median of Means (TMM) as implemented in DESeq2, and differentially expressed genes determined according to a Negative Binomial model as per DESeq2.

### Proteomic analysis of *D. gadei*

**Sample preparation.** *Dysgonomonas gadei* cells were suspended in 5% sodium dodecyl sulfate (SDS) in 50 mM triethylammonium bicarbonate (TEAB) pH 7.5. The samples were subsequently sonicated using an ultrasonic homogenizer (Hielscher) for 1 minute. The whole-cell lysate was centrifuged at 10,000 × *g* for 5 min to remove cellular debris. Protein concentration was determined using a bicinchoninic acid (BCA) protein assay (Thermo Scientific). A total of 20 μg protein was used for further processing. Proteins were reduced by incubation with 20 mM tris(2-carboxyethyl)phosphine for 15 min at 47 °C, and

subsequently alkylated with 20 mM iodoacetamide for 30 minutes at room temperature in the dark. Proteomic sample preparation was performed using the suspension trapping (S-Trap) sample preparation method[50], as recommended by the supplier (ProtiFi™, Huntington NY). Briefly, 2.5 μl of 12% phosphoric acid was added to each sample, followed by the addition of 165 μl S-Trap binding buffer (90% methanol in 100 mM TEAB pH 7.1). The acidified samples were added, separately, to S-Trap micro-spin columns and centrifuged at 4,000 × *g* for 1 min until the solution has passed through the filter. Each S-Trap micro-spin column was washed with 150 μl S-trap binding buffer by centrifugation at 4000 × *g* for 1min. This process was repeated for a total of five washes. Twenty-five μl of 50 mM TEAB containing trypsin (1:10 ratio of trypsin:protein) was added to each sample, followed by proteolytic digestion for 2 h at 47 °C using a thermomixer (Eppendorf). Peptides were eluted with 50 mM TEAB pH 8.0 and centrifugation at 1000 × *g* for 1 min. Elution steps were repeated using 0.2% formic acid and 0.2% formic acid in 50% acetonitrile, respectively. The three eluates from each sample were combined and dried using a speed-vac before storage at −80 °C.

**Mass Spectrometry.** Peptides were dissolved in 2% acetonitrile containing 0.1% trifluoroacetic acid, and each sample was independently analyzed on an Orbitrap Fusion Lumos Tribrid mass spectrometer (Thermo Fisher Scientific), connected to an UltiMate 3000 RSLCnano System (Thermo Fisher Scientific). Peptides were injected on a PepMap 100 C₁₈ LC trap column (300 μm ID × 5 mm, 5 μm, 100 Å) followed by separation on an EASY-Spray nanoLC C₁₈ column (75 μm ID × 50 cm, 2 μm, 100 Å) at a flow rate of 250 nl/min. Solvent A was water containing 0.1% formic acid, and solvent B was 80% acetonitrile containing 0.1% formic acid. The gradient used for analysis was as follows: solvent B was maintained at 2% for 5 min, followed by an increase from 2 to 35% B in 120 min, 35–90% B in 0.5 min, maintained at 90% B for 4 min, followed by a decrease to 3% in 0.5 min and equilibration at 2% for 10 min. The Orbitrap Fusion Lumos was operated in positive-ion data-dependent mode. The precursor ion scan (full scan) was performed in the Orbitrap in the range of 400–1600 *m/z* with a resolution of 120,000 at 200 *m/z*, an automatic gain control (AGC) target of $4 \times 10^5$ and an ion injection time of 50 ms. MS/MS spectra were acquired in the linear ion trap (IT) using Rapid scan mode after high-energy collisional dissociation (HCD) fragmentation. An HCD collision energy of 30% was used, the AGC target was set to $1 \times 10^4$ and dynamic injection time mode was allowed. The number of MS/MS events between full scans was determined on-the-fly to maintain a 3 s fixed duty cycle. Dynamic exclusion of ions within a ± 10 ppm *m/z* window was implemented using a 35 s exclusion duration. An electrospray voltage of 2.0 kV and capillary temperature of 275 °C, with no sheath and auxiliary gas flow, was used.

All mass spectra were analyzed using MaxQuant 1.6.12.0[51], and searched against the *Dysgonomonas gadei* ATCC BAA-286 proteome database downloaded from Uniprot (accessed 09.01.2020). Peak list generation was performed within MaxQuant and searches performed using default parameters and the built-in Andromeda search engine[52]. The enzyme specificity was set to consider fully tryptic peptides, and two missed cleavages were allowed. Oxidation of methionine, N-terminal acetylation and deamidation of asparagine and glutamine were allowed as variable modifications. Carbamidomethylation of cysteine was allowed as a fixed modification. A protein and peptide false discovery rate (FDR) of less than 1% was employed in MaxQuant. Proteins that contained similar peptides and could not be differentiated based on MS/MS analysis alone were grouped to satisfy the principles of parsimony. Reverse hits, contaminants, and proteins only identified by site modifications were removed before downstream analysis. The mass spectrometry proteomics data have been deposited to the ProteomeXchange Consortium via the PRIDE [1] partner repository with the dataset identifier PXD039984.

### Generation of expression constructs

Unless stated otherwise genes were purchased as codon-optimized constructs from Twist Biosciences. *D. gadei* and *B. finegoldii* genes were cloned from genomic DNA using standard restriction cloning methods, followed by ligation into pET28a vectors and transformation into TOP10 cells (Novagen) with subsequent sequencing of selected purified recombinant plasmids by Eurofins for confirmation.

### Protein expression and purification

*Expression and purification of $Dg_{GH185a}$, $Dg_{GH185b}$, $Phage_{GH185}$, $Myxo_{GH185}$, $Dg_{GH172a}$, $Dg_{GH172b}$, $Dg_{GH172c}$, and $Myc_{GH172}$.*

Recombinant proteins were expressed in competent *E. coli* Tuner cells (Novagen) using pET28a vectors generated as above. Cells were grown in LB media at 37 °C with shaking, until turbidity reached an $OD_{600}$ of ~0.6, wherein expression was induced with 0.2 mM IPTG and cells were further cultured for 16 hours at 16 °C. Sonication was used to lyse cells in 20 mM Tris, pH 8.0, 200 mM NaCl.

Enzymes were purified using immobilized metal affinity chromatography on cobalt TALON resin. Proteins were dialyzed into 20 mM HEPES, pH 8.0, 150 mM NaCl buffer dialysis (Medicell). For crystallography proteins were purified further via size-exclusion chromatography (HiLoad 16/600 Superdex 200, GE Healthcare) in 20 mM HEPES, pH 8.0, 150 mM NaCl. Protein purity was ascertained by SDS-PAGE and protein concentrations were determined using Nanodrop spectroscopy (Thermofisher).

### Purification of Rv3707c and MSMEG_2107

For protein Rv3707c and MSMEG_2107 expression, an aliquot of competent BL21 DE3 *Escherichia coli* was transformed with plasmid and plated on LB agar supplemented with 50 μg/mL kanamycin. One plate of bacteria was scraped to inoculate 1 L of modified Studier's autoinduction media[53]. The bacteria were incubated at 37 °C with shaking until $OD_{600}$ = ~0.6, whereupon the flasks were cooled on ice with agitation to 20 °C and then returned shake overnight at 20 °C. After induction, the cultures were pelleted at 3990 × *g* for 25 min at 4 °C. Pellets were resuspended in sterile PBS and pelleted at 7000 x *g* for 10 min. The supernatant was removed, and pellets were snap frozen in liquid nitrogen and stored at −20 °C until preparation.

Rv3707c was purified by suspension of one pellet in cold lysis buffer (25 mM HEPES pH8; 400 mM NaCl; 5% glycerol; 50 mM L-arginine; 50 mM L-glutamic acid; 1 mM beta-mercaptoethanol). 1 mg ml⁻¹ deoxyribonuclease I from bovine pancreas (Sigma-Aldrich) was added to cell slurry and incubated for 30 min. Cells were lysed by three passages through a French pressure cell. Insoluble debris was then pelleted by centrifugation at 40,000 × *g* for 40 min (4 °C). The supernatant was then processed by immobilized metal affinity chromatography (IMAC) on a gravity column containing 2 mL bed volume of cOmplete His Tag purification resin (Roche). After loading the lysate, the column was washed with 80 mL of lysis buffer, then eluted with an imidazole gradient of 50, 100, 250, and 500 mM. Protein-containing fractions were pooled and dialyzed exhaustively against three liters of dialysis buffer (25 mM HEPES pH8; 400 mM NaCl; 5% glycerol; 50 mM L-arginine; 50 mM L-glutamic acid; 2 mM dithiothreitol). The crude protein was concentrated to a final volume of 0.5 mL on a 30k Da molecular weight cutoff Pierce protein concentrator (Thermo Scientific). This fraction was then further purified by size exclusion chromatography on an AKTA Prime system with a SuperDex 26/600 S200 column in the above dialysis buffer before being concentrated. The protein was always used freshly prepared.

### Purification of $Noc_{GH172}$ and $Mab_{GH4185}$

One plate of BL21-DE3 transformed with an appropriate plasmid was used to inoculate 1 L Terrific Broth supplemented with kanamycin. The culture was grown to an $OD_{600}$ of 0.6 and induced with 0.25 mM IPTG and grown overnight at 20 °C. After harvest of biomass as in the purification of Rv3707c, cell pellets were resuspended in a buffer containing 25 mM HEPES; 40 mM NaCl, lysozyme, and DNAse I. Subsequent purification steps were identical to those in Rv3707c, but in the above, simpler buffer, omitting lysozyme and DNAse I.

### Summary of methods for phylogenetic analyses

Two alignments were used to infer respectively a global phylogeny for members of the PF13810 family and a more restricted phylogeny focusing on close relatives to the functionally characterized proteins. The global alignment was derived from the "Full" Pfam alignment for PF13810 made of 1145 sequences and 1321 aligned sites (http://pfam.xfam.org/family/PF13810#tabview=tab3). This led to an alignment of 753 sequences and 179 aligned sites by: (i) deleting partial sequences that did not include the catalytic residues or that corresponded to obsolete sequences (nine sequences) and (ii) adding the three sequences from *Mycobacterium abscessus* (strain 4529 available at the Integrated Microbial Genomes & Microbiomes (IMG/M) database: 2635695100/Ga0069448_11324, 2635694794/Ga0069448_1118, 2635698039/Ga0069448_113269) and (iii) deleting sites made of a majority of indels. For the restricted alignment 39 complete sequences were aligned, including the seven proteins investigated in this study - enzymatic characterization and one structure. The sequences were aligned with Clustal Omega using default settings in SEAVIEW v.4.6.4[54,55]. Following minor manual adjustments of the alignment the mask function of SEAVIEW was used to selected aligned residues that included conserved blocks of sequences with no more than two indels leading to 200 aligned sites. The DUF4185 alignment is available as Supplementary Dataset 1 for respectively the global and restricted alignment. IQ-TREE (v.1.6.12) was used to generate maximum likelihood phylogenies using automatic model selection[56]. The selected models were LG + F + I + G4 for the global alignment and WAG + I + G4 for the restricted alignment using the Bayesian Information Criterion (BIC). Bootstraps (100 replicates) were computed to assess branch reliability. iTOL (interactive tree of life) was used to generate the figures[57]. The global phylogeny was annotated with taxonomy information derived from the UniProt database (https://www.uniprot.org/)[58].

### Timepoint assays

To assess enzymatic activity, reactions were initiated containing substrates (in water) and enzymes (in 20 mM HEPES pH 7.5, 150 mM NaCl, unless otherwise stated) of various concentrations, with 50 mM potassium phosphate buffer pH 7.2 as a dominant reaction buffer. Reactions were incubated at 37 °C for 16 hours and subsequently boiled to ensure enzymatic cessation. Time point samples were then analyzed using TLC or IC-PAD.

### Porous graphitic carbon chromatography clean-up of Rv3707c and MSMEG_2107 arabinogalactan hydrolysis assays

Due to the high concentration of L-arginine and L-glutamic acid in the buffer used for purification of Rv3707c and MSMEG_2107, enzyme assays were unsuitable for HPAEC-PAD analysis without prior solid phase extraction. To this end, at each timepoint, reaction mixtures were loaded onto a Hypersep Hypercarb SPE cartridge (Thermo Scientific) which had been washed with acetonitrile and 50% THF in water and exhaustively equilibrated with water prior to loading. Reaction products were then eluted in 80% acetonitrile in ddH2O with 0.1% trifluoracetic acid (Sigma-Aldrich) and dried by evaporation in a SpeedVac concentrator before being reconstituted in the original volume of water.

### Kinetic analysis of GH172 arabinofuranosidase activity

Enzymes (100 nM) were incubated with the indicated concentrations of pNP-α-D-Ara*f* or AG. Assays were performed in technical triplicate at 37 °C in 20 mM HEPES pH 7.5. for pNP, absorbances were measured at

400 nm and graphs were plotted in GraphPad Prism 9.3.1. For Ag, arabinose concentration was measured by IC-PAD (see below) with reference to a standard curve.

## Thin-layer chromatography (TLC)

Purified proteins were incubated with at 0.1–5 μM (as indicated) with substrates (for methanolysis, methanol was added to the reaction mixture at a final concentration of 10%) for 16 h at 37 °C to ensure reaction completion (unless otherwise indicated). Using TLC plate aluminum foils (Silicagel 60, 20 × 20, Merck) which were cut to the desired size (minimum height of 10 cm), these reaction samples were spotted (6 μl, unless otherwise indicated) and allowed to dry. TLC plates were run (twice) in solvent (1-butanol/acetic acid/ water 2:1:1 (v/ v)). Plates were then removed and dried before visualization of sugars was obtained via immersion of TLC plate in Orcinol stain. Plates were dried and developed through heating between 50 °C and 80 °C.

## Ion Chromatography with Pulsed Amperometric Detection (IC-PAD)

Oligosaccharides from enzymatic polysaccharide digestion were analyzed using a CARBOPAC PA-300 anion exchange column (Thermo-Fisher) on an ICS-6000 system. Detection enabled by PAD using a gold working electrode and a PdH reference electrode with standard Carbo Quad waveform. Buffer A – 100 mM NaOH, Buffer B – 100 mM NaOH, 0.5 M Na Acetate. Each sample was run at a constant flow of 0.25 ml·min$^{-1}$ for 100 min using the following program after injection: 0-10 min; isocratic 100% buffer A, 10-70 min; linear gradient to 60% buffer B, 70-80 min; 100% buffer B. The column was then washed with 10 mins of 500 mM NaOH, then 10 min re-equilibration in 100% buffer A. L-arabino-oligosaccharides (DP = 2–9) obtained commercially (Megazyme) were used as standards at a concentration of 25 μM. Data were processed using Chromeleon™ Chromatography Management System V.6.8. Final graphs were created using GraphPad Prism 8.0.1.

## Purification of mycobacterial arabinogalactan

Large scale purification of mycobacterial arabinogalactan was achieved by established methodologies[59]. In brief, 8 L of mycobacterial culture was grown to mid-exponential phase, cultures were pelleted and resuspended in a minimal volume of phosphate-buffered saline (VWR) and lysed using an Emulsiflex. The lysate was then boiled in a final concentration of 1% sodium dodecyl sulphate (SDS) and refluxed. Insoluble material (containing mycolyl-arabinogalactan-peptidoglycan complex) was collected by centrifugation and washed exhaustively with water to remove SDS. The mycolate layer was removed by saponification by KOH in methanol at 37 °C for 3 days. Cell wall material was then washed repeatedly to remove saponified mycolic acids with diethyl ether. The phosphodiester linkage between AG and PG was then cleaved by treatment with $H_2SO_4$ at 95 °C before being neutralized with sodium carbonate. The resultant solubilized arabinogalactan was collected in the supernatant, dialyzed exhaustively against water and lyophilized (yield = ~22.5 mg·L$^{-1}$).

## Purification of D-arabinan

Arabinogalactan (5 mg/mL) was digested in a 10 mL total volume of 25 mM MOPS buffer pH 7. One μM final concentration of BAC-FIN_04787 and BACFIN_08810 were added, and the reaction was incubated at 37 °C for 48 h. An aliquot of the reaction was analyzed by TLC to verify the hydrolysis of the substrate and galactose release. Then, the sample was dialyzed overnight against 5 L of deionized water using a 1 kDa membrane, to eliminate residual galactose from the reaction mixture. The sample was then freeze-dried and resuspended in 0.4 mL water. A TLC analysis confirmed the elimination of the residual galactose from the sample, and this was further confirmed through acid hydrolysis of the resulting D-arabinan to determine total abundance. To quantify the purity of isolated d-arabinan, an aliquot of

both (0.25 mg/mL) d-arabinan and arabinogalactan were treated by acid hydrolysis using 300 mM HCl at 100 °C for 1 h. Samples were neutralised to pH 7 with NaOH and analysed by HPAEC. Quantitation was based on the migration of standards and the ratio between galactose and arabinose.

## Purification of mycobacterial lipoglycans

One liter of mycobacterial culture was grown to mid-exponential phase and pelleted as above. The pellet was resuspended in 20 ml PBS, 0.1% Tween-80, chilled and lysed by bead-beating. Lysate was transferred to a Teflon-capped glass tube and vortexed with an equal volume of citrate buffer saturated with phenol (Sigma-Aldrich), and heated to 80 °C for 3 h, vortexing every hour. A biphase was generated by centrifugation at 845 x g at 10 °C, and the upper aqueous phase transferred to a fresh glass tube and hot phenol wash repeated twice more. The resultant protein-free glycan mixture was dialyzed exhaustively against tap water overnight to remove trace phenol and lyophilized, yielding 34 mg of crude lipoglycans (LAM, LM, PIMS) per liter of culture.

## *Pseudomonas aeruginosa* pilin oligosaccharide extraction

Pilins were purified as described by Burrows and colleagues, with some modifications[30]. Briefly, *Pseudomonas aeruginosa* PA7 were streaked out in a grid pattern onto LB agar plates and grown for 24 hours at 37 °C. Cells were then scraped from all plates using sterile cell scrapers and resuspended in 4 ml of sterile phosphate-buffered saline (pH 7.4) per plate.

Pili were then sheared from the cell wall by vigorous vortexing of resuspended cells for 2 min. This suspension was centrifuged for 5 min at 6000 × g. The supernatant was centrifuged for 20 min at 20,000 × g. Supernatants were transferred to fresh tubes and $MgCl_2$ was added to give a final concentration of 100 mM. Following inversion of these tubes to ensure mixing, samples were incubated at 4 °C overnight, allowing precipitation of sheared proteins. Samples were then centrifuged for 20 minutes at 20,000 x g, yielding a precipitate smudge which was resuspended in 50 mM $NH_4HCO_3$, pH 8.5 and transferred to fresh Eppendorf tubes. This solution was then dialyzed into the same buffer to eliminate excess $MgCl_2$.

Bradford assays were then performed to assay the mass of protein in the sample, followed by digestion of the intact protein pilins using proteinase K in a 50:1 pilin to enzyme ratio by mass for 24 h in the presence of 2 mM $CaCl_2$. Glycans were then purified from digested proteins using porous graphitized carbon chromatography, where sugars were eluted from the column using a twofold increasing concentration series of a butan-1-ol:$H_2O$ gradient from 1:32 to 1:1 using 1 mL elutions[60]. Thin-layer chromatography (TLC) of eluates showed various oligomers of arabinan present in all fractions, all of which were subsequently used as substrates for potential arabinanases.

## Synthesis of pNP-α-D-arabinofuranoside

Para-nitrophenol (pNP)-α/β-D-arabinofuranoside synthesis was achieved following the established procedures[23].

## SEC-LS

Molecular weights were determined by size exclusion chromatography coupled light scattering using an Agilent MDS system with either an Agilent BioSEC 5 1000 Å, 4.6 × 300 mm, 5 μm or GE Superdex 200 5 15 mm columns at appropriate flow rates. Detector offsets were calibrated using a BSA standard and concentrations were determined by refractive index.

## Fluorescent-conjugated mAGP hydrolysis assay

Fluorescently labeled mycolyl-arabinogalactan-peptidoglycan complex (mAGP) was isolated from *Corynebacterium glutamicum ATCC13032* following previously reported methods[37–39]. In brief, cells were grown in the presence of 5-AzFPA to saturation, reacted with

DBCO-conjugated AF647 and then the mAGP was isolated. The isolated product was suspended in 2% SDS in PBS and split into the outlined treatment groups in Eppendorf tubes. Samples were pelleted by centrifugation at 15,000 x g for 5 min at 4 °C then washed with PBS once (100 μL). Following this wash, the mAGP was resuspended in 90 μL PBS and enzyme stock added for a final concentration of 5 μM of each enzyme. Samples were incubated at 37 °C with rotation for 16 h. Following incubation, samples were pelleted by centrifugation at 15,000x g for 5 min at 4 °C then washed with PBS twice (100 μL). The pellets were then suspended in 2% SDS in PBS, transferred to a 96-well plate and the fluorescence emission of each well was then measured on a Tecan Infinite M1000 Pro microplate reader. Monitoring of AF647 fluorescence was achieved by exciting at 648 nm ± 5 nm and detecting at 671 nm ± 5 nm. Z-position was set to 2 mm, and the fluorimeter gain was optimized and then kept constant. Data are reported in relative fluorescence units (RFU).

### Reporting summary

Further information on research design is available in the Nature Portfolio Reporting Summary linked to this article.

## Data availability

The mass spectrometry proteomics data generated in this study have been deposited in the ProteomeXchange Consortium via the PRIDE partner repository with the dataset identifier PXD039984. The RNA-seq data are available in the Sequence Read Archive database under accession code PRJNA950890. All other data generated or analyzed in this study are available within the article and its supplementary materials. Source data are provided with this paper.

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

## Acknowledgements

We thank members of the Birmingham mycobacteriology group and the Newcastle glycobiology group for support and discussions. We thank ESM and JMW for their support. This work was supported by the Biotechnology and Biological Sciences Research Council (grants BB/S010122/1 and BBSRCIAA-1544084 to PJM, BB/M011186/1 studentship to OA-J and BB/M011186/1 studentship to NPB) The Academy of Medical Sciences (SBF006\1048 to ECL, SBF005\1065 to AC), the Royal Society (RGS\R2\202228 to ECL and RGS\R2\212050 to AC) the Wellcome Trust (209437/Z/17/Z to ALL, studentship to STB), the Australian Research Council (DP210100233, DO210100235 to SJW). the European Research Council (322820 to Harry J Gilbert, supporting JM-M) the Novo-Nordisk Foundation (NNF20SA0067193, NNF22SA0077601 and NNF22OC0077058 to BH), the National Institute of Allergy and Infectious Disease (R01 Al-126592 to LK) and the NIH Common Fund (U01GM125288 to LK).

## Author contributions

O.A.-J. – Methodology, validation, formal analysis, investigation, data curation, visualization. S.B. – Methodology, validation, formal analysis, investigation, data curation, visualization, resources. J.R. – Methodology, validation, formal analysis, investigation, data curation, writing – original draft preparation, writing – review and editing. A.La. – Methodology, validation, resources. P.P. – Methodology, validation, formal analysis, resources. V.M. – Methodology, formal analysis, investigation, visualization, writing – review and editing. N.P.B. – Methodology, validation, formal analysis, visualization. T.H. – Methodology, formal analysis, resources. J.M. – Investigation. F.M. – Investigation. A.F. – Investigation, resources. J.A.-R. – Investigation. S.L.O. – Investigation. L.P. – Investigation. A.C. – Resources, supervision. G.S.A.W. – Resources, supervision. A.B. – Methodology, validation, writing – review and editing. M.T. – Resources, supervision. B.H. – Resources. J.M.-M. – Investigation. R.P.H. – Investigation, resources, writing – review and editing, supervision. L.L.K. – Resources, supervision, funding acquisition, writing – review and editing. A.Lo. – Methodology, validation, formal analysis, writing – review and editing, supervision, funding acquisition. S.J.W. – Resources, validation, formal analysis, writing – review and editing, supervision, funding acquisition. E.C.L. – Conceptualization, methodology, validation, formal analysis, data curation, writing – original draft preparation,

writing – review and editing, visualization, supervision, funding acquisition. P.J.M. – Conceptualization, methodology, validation, formal analysis, data curation, writing – original draft preparation, writing – review and editing, visualization, supervision, funding acquisition.

## Competing interests

Drs. Moynihan and Lowe are co-inventors on an unpublished patent application pertaining to some of the enzymes described in this manuscript. The remaining authors declare no competing interests.

## Additional information

[1]Newcastle University Biosciences Institute, Medical School, Newcastle University, Newcastle upon Tyne NE2 4HH, UK. [2]Institute of Microbiology and Infection, School of Biosciences, University of Birmingham, Birmingham B15 2TT, UK. [3]School of Chemistry and Bio21 Molecular Science and Biotechnology Institute, University of Melbourne, Parkville, Victoria 3010, Australia. [4]Department of Chemistry, Massachusetts Institute of Technology, Cambridge, MA, USA. [5]The Broad Institute of Harvard and MIT, Cambridge, MA, USA The Koch Integrative Cancer Research Institute, Massachusetts Institute of Technology, Cambridge, MA, USA. [6]Department of Biochemistry and Systems Biology, Institute of Systems, Molecular and Integrative Biology, University of Liverpool, Liverpool, UK. [7]School of Life Sciences, University of Essex, Colchester, UK. [8]Department of Biological Sciences, King Abdulaziz University, Jeddah, Saudi Arabia. [9]Department of Biotechnology and Biomedicine (DTU Bioengineering), Technical University of Denmark, 2800 Kgs Lyngby, Denmark. [10]Microbial Enzymology Group, Department of Applied Sciences, Northumbria University, Newcastle upon Tyne, UK. [11]These authors contributed equally: Omar Al-Jourani, Samuel T. Benedict, Jennifer Ross. ✉e-mail: elisabeth.lowe@ncl.ac.uk; p.j.moynihan@bham.ac.uk

