## [Peer Review File · Nature Communications]

Identification of D-arabinan-degrading enzymes in mycobacteriaReviewer #1 (Remarks to the Author):

This is a very nice manuscript, reporting new families of enzymes that break down D-arabinan containing polymers (from mycobacteria). The approach used to identify these proteins, using gut microbiota and bioinformatics, is novel and should serve as a general strategy for the identification of other CAZymes in the future. This is an excellent contribution to Nature Communications and I recommend publication.

I have two substantive criticisms. One relates the X-ray crystallographic work, which comes across as rather superficial and provided not much information beyond the oligomeric state of the proteins. Were attempts made to get substrate co-crystals with mutants where the key catalytic residues were knocked out? Or with potential inhibitors? If so and it failed, that should be mentioned. If it was not tried, that should also be mentioned. More broadly than that, I am not sure the structural work adds much; indeed is a bit distracting as its quite descriptive and not a lot of insights are learned. Mechanistically, these proteins seem to behave like every other GH. In addition, the DUF4185 structural work is simply modelling. The second point relates to the release of fluorescently-labeled materials from the cell wall. This is an unnecessary detour from the main messages of the paper. Removing it (and the crystallographic work) would tighten up the message and make the paper easier to read.

The authors should also consider the following minor points.

Lines 64–5. This phrase is not supported by the data in the paper “suggesting that the ability to degrade mycobacterial glycans plays an important role in the biology of diverse organisms.” What does seem to be true is that the ability to degrade D-arabinan polymers is important to diverse organisms.

Line 118. Is the use of ‘our’ here correct?

Line 166–167. Do the colors of the filled circles have any meaning? If so, that should be defined. If not, it would be better just to use black/gray (as was done in a later figure). Or use the conventional +/-.

Line 193-4. Can the authors speculate on the significance that fucose isomerase was the most abundant CAZyme? It is not obvious to this reviewer.

Line 243. “Araf,” not “araF”

Line 243–4. This is not a true statement: “The D-arabinan branches of lipoarabinomannan (LAM) are structurally conserved with those in AG.” The two structures are similar, but conserved implies much greater similarity than exists. The AG arabinan is structurally well defined; LAM arabinan is much more heterogenous, with few distinct motifs.

Lines 435–6 This sentence should be fixed: “Given that branched D-arabinan is the only conserved structure within both AG and LAM,” See comment above.

Lines 528–537. This paragraph seems like a detour given that the paper largely ignores the galactofuranosidase activities identified, except to use the proteins to ‘clean-up’ AG. Likewise, the reference to chain length at the end of the paragraph is tangential to the story. This could be shortened or removed.

One final comment:

It would have been easier to review this manuscript had page numbers, in addition, to line numbers ,been present.

Reviewer #2 (Remarks to the Author):

The manuscript entitled "Mining the human gut microbiome identifies mycobacterial D-arabinan degrading enzymes" by Al-Jourani and co-workers disclosed glycolytic capacity of the human gut microbiota and discovered a new member of the DUF2961 family (GH172), and a novel family of enzymes able to degrade mycobacterial arabinogalactan. As pointed out by the authors in the introduction, no enzymes have been characterized with either exo- or endo- D-arabinofuranase activity (enzymes that cut within the D-arabinan polymers) against AG, even when this activity was described in protein extracts collected from soil bacteria. The authors show that gut microbiome can produce enzymes that depolymerize mycobacterial D-arabinan and D-galactan through the demonstration of growth on free galactose and arabinose. The authors also show that DUF2961 family had glycoside hydrolase activity on both mycobacterial AG and LAM. Through the combination of experiments like Ion chromatography with pulsed amperometric detection (IC-PAD), it was demonstrated that GH172 enzymes are specific for α -D-arabinofuranoside linkages, and are active on D-arabinan, which can also cleave α -1,5- D-arabinofuranoside.

The manuscript seems long and difficult to follow in places. I don't think it will be of general interest and would be better in a more specialized journal, and possibly split into two or more papers. The authors may wish to address the following comments before they submit to a new journal.

Abstract Line 59: *D. gadei* as a D-arabinan degrader. Specify the genus.

Line 137 should have a comma removed at the end, and in Line 138 a comma should be after mycobacteria.

Figure 1 might benefit from including another image of PAD response of *D. gadei*, arabinose, and galactose. It is difficult to distinguish colors and identify the peak line showing galactose in *D. gadei*.

The manuscript generates confusion about the ligands/substrates binding sites and their mode of binding. Two calcium ions were observed close to the active site (Fig. 3E, green sphere), although the buffers used during protein purification and crystallization contained no divalent metal ions. For example, the authors do not mention the source of calcium bound to the ligand bound structure. It would be more interesting to the readers if they make a comment on the significance of two calcium ions within the binding pocket

Line 286: Seven calcium ions are coordinated within each Dg GH172c trimer: two within each active site/protomer, and one in the 3-fold axes.

Note: it would be useful if the authors labeled the figures with hydrogen bonding distances
How the metal ion in the crystal structure was assigned as Ca²⁺?

It is confusing as to which state (active or inactive, or both?) have been obtained in the reported crystals. It would be interesting to determine whether the calcium ion has a specific role in the activity of Dg GH172c. It would be also worthwhile that assays need to be performed in the presence of EGTA, a chelating agent that binds Ca²⁺ with a significantly larger affinity than EDTA does.

Line 319: Authors claimed that their data support assigning residues E254 and E233 as catalytic residues (acid/base or nucleophile) and highlight an important role for the adjacent D255 residue. In the Dg GH172c catalysis study, the authors have assigned E233 as catalytic residue and the same residue coordinate with Ca²⁺ binding. Is the main-chain carbonyl of the catalytic acid/base residue (E233) is involved in the calcium coordination? If so, calcium appears to be important for the enzymatic mechanism of the enzyme, probably by directly influencing the protonation state of the catalytic carboxylate.

Line 336: Might be interesting to attempt to get crystal diffraction data of the E233A variant to determine if Ca binding is there

Line 498-502: Mention of "significant structural flexibility" in regard to Rv3707c. Maybe addition of Ca²⁺ to stabilize the protein and then obtain diffraction data.

This needs to be discussed and put in relation to activity data.

Previous studies have shown that for GHs, Ca²⁺ often contribute to protein stabilization, as exemplified by GH13 α -amylases Ban, X., et al; (2021)- and well as several GHs use a Ca²⁺ for

substrate recognition: for example, GH43 α -L-arabinanase, De Sanctis, et al; (2010).

The results section would benefit from more descriptive, shorter subtitles. For example, line 192 could be "Identification of D-arabinan degradation PUL".

Figure 7 is fantastic and really helps readers understand all the information in one concise image.

Nature Communications- Reviewer comment questionnaire

What are the noteworthy results?

- 12 Bacteroidetes species found in human gut were able to grow off arabinogalactan (AG), a unique polysaccharide to mycobacterial
- Exploited natural processes to produce pure D-arabinan by increasing PLUs encoding for glycoside hydrolase
- When growing *D.gadei* using D-arabinan, there were elevated levels of fucose isomerase and other proteins from PUL42 group; the study prioritizes enzymes that have homology to unknown proteins in PUL42
- It is my opinion the most important results come from line 441 and on, when they determine if the enzyme can cleave AG in a cell wall ; the AG is fluorescently labeled and if any material is released from the cell wall, it will show via the release of a fluorescent product
- Interesting to read about the evolutionary conservation and how the authors deduced that some enzyme to degrade AG must exist in our gut
- Found enzymes that can degrade Mycobacterial AG

Will the work be of significance to the field and related fields? How does it compare to the established literature? If the work is not original, please provide relevant references.

- Yes ; these enzymes focus on degrading parts of the cell wall that are necessary and only in mycobacteria; by focusing on these enzymes and creating drug compounds to inhibit them, this can result in lethal consequences of the bacteria
- Some of the work supports outside reports: Line 262 supports outside claims relating to oligomeric states –

Does the work support the conclusions and claims, or is additional evidence needed?

- overall the work supports the claims that there are bacteria in the human gut capable of degrading AG (see figure 1B)
- Line 314 about conserving glutamate residues is supported by lines 316-318, about mutagenizing the carboxylate to alanine and generating several variants with no activity –

Are there any flaws in the data analysis, interpretation, and conclusions? - Do these prohibit the publication or require revision?

- It took me several times of rereading to understand the protein names; a more simple explanation followed immediately by the PUL schematic would make it easier to understand .
- Why did we use D-araF containing Pilin oligosaccharides in line 243?; is this for a binding assay
- I'm also confused about how exactly they picked the 14 strains at the beginning of the methods section in line 621-626

Is the methodology sound? Does the work meet the expected standards in your field?

- Yes, methods are thorough and detailed. Include lots of reasoning in the results section –

Is there enough detail provided in the methods for the work to be reproduced?

- Yes, there is great explanation in the methods to replicate the work done

Reviewer #3 (Remarks to the Author):

In this article, using a smart approach with selected Bacteroidetes strains, the authors discovered several enzymes able to degrade the arabinogalactan polymer (AG) of Corynebacteriales cell

envelope. Importantly, only a single enzyme degrading AG was described before this work. Al-Jourani et al. identified and characterized a completely new family of glycoside hydrolases and determined their structures. They also identified new members of the DUF2961 family and characterized their activity. This is a very exhaustive work that adequately support the conclusions drawn by the authors.

Arabinogalactan is a polymer that is a hallmark of Mycobacteria. Although enzymes involved in its assembly are known, the exact steps of its biogenesis as a function of time and space are very poorly documented. In this respect, the identification of endo and exo enzymes able to cleave AG in Mycobacteria will greatly help to understand the synthesis of AG during cell elongation and division. This is a major breakthrough in the field and will undoubtedly help to unravel the very atypical mechanism of division of Mycobacteria and related species.

I have minor comments/questions:

- In Figure 1E, a quite small amount of galactose is detected in lane 3. What is the major product detected just under galactose ?

- Table S1 is not present in the document ? May be it has been deleted during pdf conversion ?

- Proteomics studies of *D. gadei*: it is a whole cell proteomic study. This is surprising since a secretome study would have been more appropriate and much simpler to analyze ? Is there any specific reason that I didn't understand ? In that case, may be useful to justify the choice.

- Fig S3: why arabinose is not migrating exactly like "reference" arabinose ?

- I don't understand from the article if orthologs of DUF2961 are also found outside the order of Corynebacteriales ?

- Figure S16: The mycolate chains are very hydrophobic. I guess your mAGP preparation is not soluble in PBS without SDS making very difficult to perform and interpret this enzymatic assay on "intact" cell wall.

What is the real observed activity (not relative results) ?

And how it compares to your activity on purified arabinogalactan polymers ?

May be it is useful to show the negative control, without enzyme and /or with inactive enzymes (with substituted catalytic amino acids).

Nicolas Bayan

Response to reviewers

We would like to thank the reviewers for their constructive feedback on our manuscript and careful reading of the text. We have followed the Editor's suggestion of removing the structural biology data from the manuscript, which we will prepare for a future publication. We are grateful for this suggestion as we believe it improves the readability of the paper overall. Our specific responses to the reviewer's comments are found below. Numbers in parenthesis reflect the new line numbers in the revised manuscript.

Reviewer 1:

One relates the X-ray crystallographic work, which comes across as rather superficial and provided not much information beyond the oligomeric state of the proteins. Were attempts made to get substrate co-crystals with mutants where the key catalytic residues were knocked out? Or with potential inhibitors? If so and it failed, that should be mentioned. If it was not tried, that should also be mentioned. More broadly than that, I am not sure the structural work adds much; indeed is a bit distracting as its quite descriptive and not a lot of insights are learned. Mechanistically, these proteins seem to behave like every other GH. In addition, the DUF4185 structural work is simply modelling.

Following advice from other reviewers and Editor, we have removed all structural data from the paper making these comments no longer applicable to the manuscript. We will consider these points for future manuscript(s) that includes our experimentally determined structures of both the DUF4185 enzyme and the GH172 enzyme.

The second point relates to the release of fluorescently-labeled materials from the cell wall. This is an unnecessary detour from the main messages of the paper. Removing it (and the crystallographic work) would tighten up the message and make the paper easier to read.

We acknowledge that these data were not adequately explained in the text, and in response to this and Reviewer 3's comments have revised the depiction of the data and its explanation. Please see below for a more detailed description of the changes (Figure S13, Line 410-418).

Lines 64–5. This phrase is not supported by the data in the paper “suggesting that the ability to degrade mycobacterial glycans plays an important role in the biology of diverse organisms.” What does seem to be true is that the ability to degrade D-arabinan polymers is important to diverse organisms.

We have revised to reflect the Reviewer's suggestion. (Line: 64)

Line 118. Is the use of 'our' here correct.

We have reworded this sentence for clarity. (Line 118)

Line 166–167. Do the colors of the filled circles have any meaning? If so, that should be defined. If not, it would be better just to use black/gray (as was done in a later figure). Or use the conventional +/-.

We have recoloured the filled circles to grey as suggested. (Line 158).

Line 193-4. Can the authors speculate on the significance that fucose isomerase was the most abundant CAZyme? It is not obvious to this reviewer.

In this system the bacteria are reliant upon AG as a sole carbon source. While there are multiple glycoside hydrolases to generate monosaccharides for carbon use, there is only one fucose isomerase in this PUL. The isomerisation step is likely required for utilisation of the arabinose, and so it is a likely bottleneck for carbon acquisition in this context. As such, we anticipate that the bacteria respond by producing more of this protein. However, detailed investigation of the PUL expression dynamics is beyond the scope of this manuscript. (Line 194)

Line 243. “Araf,” not “araF”.

We thank the reviewer for noticing this error and have corrected the text in this location and throughout. (Line 249)

Line 243–4. This is not a true statement: “The D-arabinan branches of lipoarabinomannan (LAM) are structurally conserved with those in AG.” The two structures are similar, but conserved implies much greater similarity than exists. The AG arabinan is structurally well defined; LAM arabinan is much more heterogenous, with few distinct motifs.

Our use of the word conserved was in reference to the types of linkages broadly found between the residues, however we realise now that this was unclear and thank the reviewer for their suggestion. We have re-worded this sentence to say that ““The D-arabinan branches of lipoarabinomannan (LAM) are structurally similar to those in AG.” (Line 250)

Lines 435–6 This sentence should be fixed: “Given that branched D-arabinan is the only conserved structure within both AG and LAM,” See comment above.

As above, we have tried to edit for clarity “Given the presence of D-arabinan motifs in both AG and LAM,” (Line 402)

Lines 528–537. This paragraph seems like a detour given that the paper largely ignores the galactofuranosidase activities identified, except to use the proteins to ‘clean-up’ AG. Likewise, the reference to chain length at the end of the paragraph is tangential to the story. This could be shortened or removed.

While we understand the reviewer’s concerns, we have chosen to keep this paragraph in the manuscript. We have substantially shortened the text elsewhere and feel that some context for these enzymes is helpful for the readers. Inclusion of this discussion also reflects the sorts of questions we have received when we have presented these data. (Line 458)

It would have been easier to review this manuscript had page numbers, in addition, to line numbers, been present.

We have added page numbers to the text, our apologies for the oversight.

Reviewer 2:

Abstract Line 59: *D. gadei* as a D-arabinan degrader. Specify the genus.

This has been amended. (Line 59)

Line 137 should have a comma removed at the end, and in Line 138 a comma should be after mycobacteria.

This has been changed. (Line 136)

Figure 1 might benefit from including another image of PAD response of *D. gadei*, arabinose, and galactose. It is difficult to distinguish colors and identify the peak line showing galactose in *D. gadei*.

This has been added as an inset panel. (Figure 1, Line 158)

The manuscript generates confusion about the ligands/substrates binding sites and their mode of binding. Two calcium ions were observed close to the active site (Fig. 3E, green sphere), although the buffers used during protein purification and crystallization contained no divalent metal ions. For example, the authors do not mention the source of calcium bound to the ligand bound structure. It would be more interesting to the readers if they make a comment on the significance of two calcium ions within the binding pocket

Line 286: Seven calcium ions are coordinated within each Dg GH172c trimer: two within each active site/protomer, and one in the 3-fold axes.

Note: it would be useful if the authors labeled the figures with hydrogen bonding distances
How the metal ion in the crystal structure was assigned as Ca²⁺?

It is confusing as to which state (active or inactive, or both?) have been obtained in the reported crystals. It would be interesting to determine whether the calcium ion has a specific role in the activity of Dg GH172c. It would be also worthwhile that assays need to be performed in the presence of EGTA, a chelating agent that binds Ca²⁺ with a significantly larger affinity than EDTA does.

Line 319: Authors claimed that their data support assigning residues E254 and E233 as catalytic residues (acid/base or nucleophile) and highlight an important role for the adjacent D255 residue.

In the Dg GH172c catalysis study, the authors have assigned E233 as catalytic residue and the same residue coordinate with Ca²⁺ binding. Is the main-chain carbonyl of the catalytic acid/base residue (E233) is involved in the calcium coordination? If so, calcium appears to be important for the enzymatic mechanism of the enzyme, probably by directly influencing the protonation state of the catalytic carboxylate.

Line 336: Might be interesting to attempt to get crystal diffraction data of the E233A variant to determine if Ca binding is there

Line 498-502: Mention of "significant structural flexibility" in regard to Rv3707c. Maybe addition of Ca²⁺ to stabilize the protein and then obtain diffraction data.

This needs to be discussed and put in relation to activity data.

Previous studies have shown that for GHs, Ca²⁺ often contribute to protein stabilization, as exemplified by GH13 α -amylases Ban, X., et al; (2021)- and well as several GHs use a Ca²⁺ for substrate recognition: for example, GH43 α -L-arabinanase, De Sanctis, et al; (2010).

As stated above, we have removed all structural data from the paper making these comments no longer applicable to the manuscript. We thank the reviewer for the helpful suggestions, which will be considered as we build another manuscript with our structural data.

The results section would benefit from more descriptive, shorter subtitles. For example, line 192 could be "Identification of D-arabinan degradation PUL".

We have amended the heading per the reviewer's suggestion (Line 188) as well as the heading at line 392.

Figure 7 is fantastic and really helps readers understand all the information in one concise image.

We thank reviewer 2 for this comment. Considering the removal of the structural data, we have edited this slightly to reflect this change. (Figure 6, Line 440)

It took me several times of rereading to understand the protein names; a more simple explanation followed immediately by the PUL schematic would make it easier to understand.

We appreciate that the complexity of having multiple PULs with multiple gene names can be challenging. This is part of our motivation for Figure 6 (was Figure 7) in the manuscript, so that the reader could quickly refer to the most important activities and gene names.

Why did we use D-araF containing Pilin oligosaccharides in line 243?; is this for a binding assay.

The pilin oligosaccharides are formed of α -1,5 linkages exclusively, which is the same as the main backbone in AG and LAM. This allowed us to investigate linkage specificity of the enzymes without requiring access to synthetic substrates. We have re-worded this section slightly to reflect this. (Line 243-244)

I'm also confused about how exactly they picked the 14 strains at the beginning of the methods section in line 621-626.

The strains were chosen as they are type strains representative of the major genera of Bacteroidetes, which were readily cultured in our hands. (Line 548)

Reviewer 3:

In Figure 1E, a quite small amount of galactose is detected in lane 3. What is the major product detected just under galactose?

The major product is galactobiose as the enzyme is endo-acting. A more detailed explanation of this was not included as this is not the focus of our manuscript. (Line 158, Figure 1)

Table S1 is not present in the document? May be it has been deleted during pdf conversion?

Due to its length, Table S1 was provided as a separate Microsoft Excel file. This was provided to the journal at submission. We have updated the Supplemental Figure and Tables file to direct the reader to this file to improve clarity.

Proteomics studies of *D. gadei*: it is a whole cell proteomic study. This is surprising since a secretome study would have been more appropriate and much simpler to analyze? Is there any specific reason that I didn't understand? In that case, may be useful to justify the choice.

Bacteroidetes generally do not secrete soluble enzymes for glycan degradation, exported enzymes are attached to the outer leaflet of the outer membrane as lipoproteins, therefore whole cell proteomics ensured we were able to detect as many relevant proteins as possible.

Fig S3: why arabinose is not migrating exactly like "reference" arabinose?

Our reference standards are typically in water whereas our assay reactions have additional salt, enzymes, etc. which influences their running on TLC. The same assays have been analysed by HPAEC-PAD (Figure 2; Line 236) which clarifies that it is arabinose in both the enzyme assays and standards.

I don't understand from the article if orthologs of DUF2961 are also found outside the order of Corynebacteriales ?

We apologise for any confusion. The gut microbes we used in our screen are all in the Bacteroidetes order, many of whom possess at least one GH172, often many more. Given that at least two activities are known for this family, we felt a phylogenetic analysis in the absence of further experimental validation would not be informative. Work is ongoing in our laboratories to further delineate this large family. Further details can be found at the CAZy website: <http://www.cazy.org/GH172.html>.

The mycolate chains are very hydrophobic. I guess your mAGP preparation is not soluble in PBS without SDS making very difficult to perform and interpret this enzymatic assay on "intact" cell wall.

We appreciate this comment. Any study of intact bacterial cell walls involves compromise given the challenging nature of these materials. We isolated and used the mycolyl-arabinogalactan-peptidoglycan (mAGP) because we wanted to determine whether the enzymes could process this physiological substrate. For example, one of the limitations of working with isolated AG is that the chains will no longer be physically capped by mycolic acids which could reasonably be expected to impact enzyme activity. Our assay is inspired by the peptidoglycan research field, where following digestion of insoluble material with conjugated fluorophores is standard. See PMID 33660879 for example.

What is the real observed activity (not relative results)?

We now include the raw data to address this question (Figure S13). We had initially included relative results for ease of interpretation for the reader but understand the merit of this point and have revised this figure accordingly.

And how it compares to your activity on purified arabinogalactan polymers ?

We did see some differences compared in the purified AG, as was expected. The full mAGP may have differences in accessibility of the linkages (and even the nature of some linkages) compared to purified AG polymers. The range of differences will depend on the specificity of the enzyme. We have reworded the text to better reflect this (Lines 410-418).

Maybe it is useful to show the negative control, without enzyme and /or with inactive enzymes (with substituted catalytic amino acids).

We agree. We modified this figure to add the negative control (without enzyme) as a data point rather than using those values to calculate the relative results that we had initially reported (Figure S13).